# Small Hydropower Assessment of Uganda Based on Multisource Geospatial Data

Petras Punys *, Gitana Vyčienė, Linas Jurevičius and Algis Kvaraciejus

Water Engineering Department, Faculty of Engineering, Vytautas Magnus University, Universiteto g. 10, LT–53361 Kaunas, Lithuania; gitana.vyciene@vdu.lt (G.V.); linas.jurevicius@vdu.lt (L.J.); algis.kvaraciejus@vdu.lt (A.K.)
* Correspondence: petras.punys@vdu.lt; Tel.: +370-37-752337

**Abstract:** This article is based on the freely available data of the web-based hydropower map HYPOSO, which the authors compiled. Only the Ugandan river network and associated hydropower potential are highlighted here, using freely available geospatial datasets. The main objective was to assess Ugandan river and stream hydropower potential, compare it with previous assessments, and identify potential sites for small hydropower plant installation. GIS techniques were extensively used to analyse hydrological and other related geospatial data. The stream-reach power potential was determined based on channel slope, the length between tributaries, and the average flow derived from a specific runoff distribution map. Stream profiles extracted from the river network's digital elevation model were validated against previous assessments. Uganda's hydropower potential was determined in various patterns, and its values were compared with prior estimates. Around 500 potential high-energy intensity stream reaches and new potential areas for small hydropower plant development were identified in this country, considering a range of characteristics. Statistical datasets were analysed, and their straightforward summaries were presented. These summary characteristics of hydropower potential are necessary for decision-makers to foster hydropower development in this country.

**Keywords:** HYPOSO map dataset; Ugandan small hydropower potential; GIS analysis; stream-reach capacity; potential hydropower sites

## 1. Introduction

### 1.1. Background

Strategic development of hydropower resources in most countries has been constrained by economic conditions and a lack of information on the river flow, topography, environmentally sensitive areas, power grid lines, and hydropower potential, especially in the African river systems [1–5]. Assessing small hydropower (SHP) sites, usually defined by an upper limit of installed capacity (P) of the plant and the country's hydropower regulations (less than 10 or 20 MW, or in some cases more), for development represents a relatively high proportion of the overall project costs. Moreover, hydropower sites are often located in remote areas with limited access to engineering teams. Therefore, high levels of experience and expertise, including tools for remote assessments, are required to conduct this assessment accurately.

The use of Geographical Information Systems (GIS) to evaluate hydropower potential has evolved over many years, with continuous advancements in computer processing capabilities leading to improvements in GIS software, which can now better handle the extensive, higher-resolution terrain data (DEM–Digital Elevation Model) that are available [6–9]. However, spatial and other related inaccuracies of these assessments cannot be entirely avoided due to the nature of the input geospatial data.

Many studies worldwide have assessed the potential of hydropower using GIS-based techniques, but few have exclusively focused on small hydro assessment in Africa [10,11].

In some parts of the African continent or specific countries, hydropower resource datasets or databases resulting from open access GIS-based hydropower mapping viewers are available [12–15]. They are available from online platforms with stream-reach capacities and individual site locations with various key datasets, and include energy, hydrology, environmental, and economic parameters. Recently, a newly developed open-access African hydropower atlas (AHA) with a database containing seasonal hydropower generation profiles for nearly all existing and several hundred future hydropower plants on the African continent was developed [16]. However, small hydropower was not covered in the database of this atlas.

Uganda currently has a limited dataset of hydropower sites but has no comprehensive hydropower database. Attempts to establish such a database based on GIS application for hydropower assessments in the country have been made [3,17,18]. A geospatial assessment of small-scale hydropower potential in Sub-Saharan Africa, including Uganda, was carried out [11]. The EU-funded project "Hydropower solutions for developing countries" (HYPOSO) has recently launched a web-based hydropower atlas for three Latin American and two African countries, namely Cameroon and Uganda [19]. It is expected to significantly improve the country's hydropower development database and provide valuable information for investing in small and medium projects.

### 1.2. The Hydropower Sector and Potential in Uganda

Uganda is a landlocked country in East Africa (Figure 1). According to the Köppen–Geiger climate classification, the majority of the land area of Uganda has a tropical savanna climate, and there is a tropical rainforest climate in the area surrounding Lake Victoria. In the northeast, Uganda is semi-arid. The greater part of Uganda consists of a plateau of 800 to 2000 m a.s.l. in height. Along the western border, in the Ruwenzori Mountains, Margherita Peak reaches an elevation of 5109 m, while on the eastern frontier, Mount Elgon rises to 4321 m a.s.l. Uganda lies almost entirely within the Nile river basin (downstream of Lake Victoria).

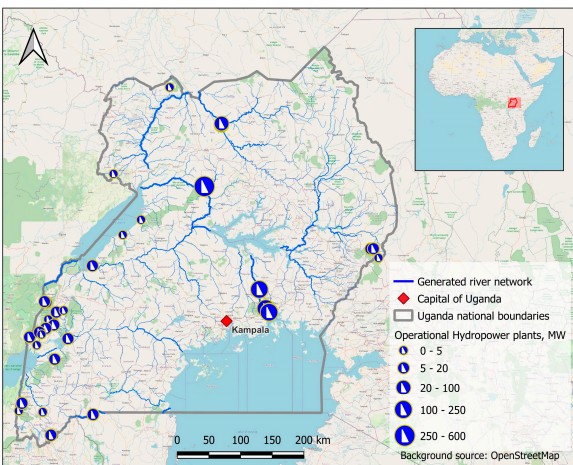

**Figure 1.** Ugandan river network and hydropower plants under operation.

Despite poor access to electricity services, electricity generation in Uganda in 2021 totalled 4749 GWh, with a clear dominance of hydropower [20]. The renewable energy potential, applications, and development in Uganda were analysed; however, hydropower potential was not detailed [21,22]. A brief profile of hydropower in Uganda has been developed [23]. This provides an overview of the power sector and renewable electricity policy and describes the hydropower sector and its potential, focusing primarily on small hydropower (SHP) policy and market analysis, education, and research on hydropower in the country. A recent overview of the hydropower sector in Uganda was provided by [5]. The African hydropower database for Uganda provides a list of operational hydropower

plants, including those under development and the one potential hydropower site [24]. In Uganda, there is a publicly accessible database of electricity generation sites containing about 150 potential hydropower locations with defined preliminary capacities [25].

As of 2021 [20], 31 hydropower plants were in operation in Uganda, with a total installed capacity of 1073 MW. The annual power generation is in the range of 4 to 5 TWh. Most of these plants and their key metrics are displayed on the HYPOSO map [19].

The gross theoretical hydropower potential of the country has not been fully assessed [26]. Table 1 summarises the hydropower potential assessments conducted in the past.

**Table 1.** Hydropower potential assessments.

| References | Technical Potential | |
| :---: | :---: | :---: |
| | GWh | MW |
| World Atlas of Hydropower & Dams [26] | 20,833 [1] | 6950 [2] |
| E. Jjunju [3], Å. Killingtveit [27] | 103,000–114,000 | 22,190–24,622 |
| JICA [28] | | 2000 [3] |
| NPA [29], NRFC [30], V. Katutsi et al. [5] | | 4137–4500 |

[1] Technically feasible potential. The economically feasible potential is 12,500 GWh. [2] Approximate. [3] Mainly the hydropower potential of the Nile.

As can be seen from this table, the magnitudes of the hydropower potential vary greatly; their accurate comparison is difficult because different approaches were used for their assessment, resulting in significant discrepancies and inconsistencies between the data and collection methods. The *International Journal on Hydropower & Dams* fundamentally defines the latter as the portion of the gross theoretical potential that could be exploited within the limits of current technology (this should include output from the currently installed capacity). Most of the above assessments include the Nile as a large hydro resource. The technical hydropower potential for Uganda identified with the GIS application HydroSearch was 103,000–114,000 GWh, which is five to six times greater than the amount given in the *World Atlas of Hydropower & Dams* [26], 20,833 GWh [27].

Hydropower is a key component in electricity generation expansion, in line with the Uganda Vision 2040 strategy [29]. About 10% of the technically feasible potential has been developed so far [31]. On average, over the past five years, hydropower plants have contributed about 90.6% of the electricity to the national utility grid [5]. An approximate share for large and small hydropower plants was 80.7 and 9.9%, respectively.

Thus far, SHP potential has not been fully assessed in this country; only rough estimates can be provided (Table 2).

**Table 2.** Small hydro (*p* < 10 MW) potential assessment.

| References | Potential, MW | | Installed Capacity, MW |
| :---: | :---: | :---: | :---: |
| | MW | GWh | |
| World Small Hydropower Development Report (WSHDR) [32] | 200 [1] | | 52 |
| World Atlas of Hydropower & Dams [26] | | >400 | 110 |
| A. Korkovelos et al. [11] | 49.8 [2] | | |

[1] The potential for SHP capacity limit <20 MW is a bit higher, 258 MW. [2] Technical potential.

In Uganda, small hydropower is generally defined as hydropower plants with an installed capacity of up to 20 MW [20]. These sites are located mainly in the country's western and eastern regions, which are hilly and mountainous.

The main aim of the present research was to evaluate small hydropower potential in Uganda, emphasising the georeferenced SHP potential site locations with their key characteristics based on the open-access information and datasets of the HYPOSO map viewer In addition, previous studies completely ignored the share of the water power potential of small-sized streams (first and second order according to the Strahler stream ordering system) to the total hydropower potential [10,11].The specific objectives of this study were as follows:

- To review the Ugandan hydropower situation and estimates of hydropower potential in the country;
- To validate the DEM for assessing stream capacities by comparing generated stream longitudinal profiles and present delineated small subbasins (catchments);
- To evaluate the country's hydropower potential compared with prior estimates;
- To identify potential site locations with their key datasets, taking into account expected capacity, protected areas, the proximity of the grid and settlements, and the energy demand concentration points, and to carry out statistical analysis.

## 2. Materials and Methods

This study focused on Ugandan small and medium-sized rivers where SHPs could be installed, excluding the Nile. The findings of the HYPOSO project, with its core product, the HYPOSO map viewer, were extensively used in this paper [19]. Exploring it or downloading the geodata sets in KML or Shape format is possible. However, the modelled estimates do not represent the actual numbers for engineering design. They provide the basis for follow-up studies to proceed with pre-feasibility or feasibility studies. The mapping methodology, GIS modelling, hydrological analysis, and use of the map viewer here were not detailed; they were only used in a general context. Large hydro was not considered in detail either.

The front page of the HYPOSO map viewer is shown in Figure 2, left. Available geospatial datasets can be explored and visualised by zooming, panning, and clicking on the map layers or icons to open the legend of this map. The interface of the map also allows for the identification of feature attributes. The map comprises 20 layers broken into five groups (Figure 2R). They are visualised once the map has been opened (Figure 2L).

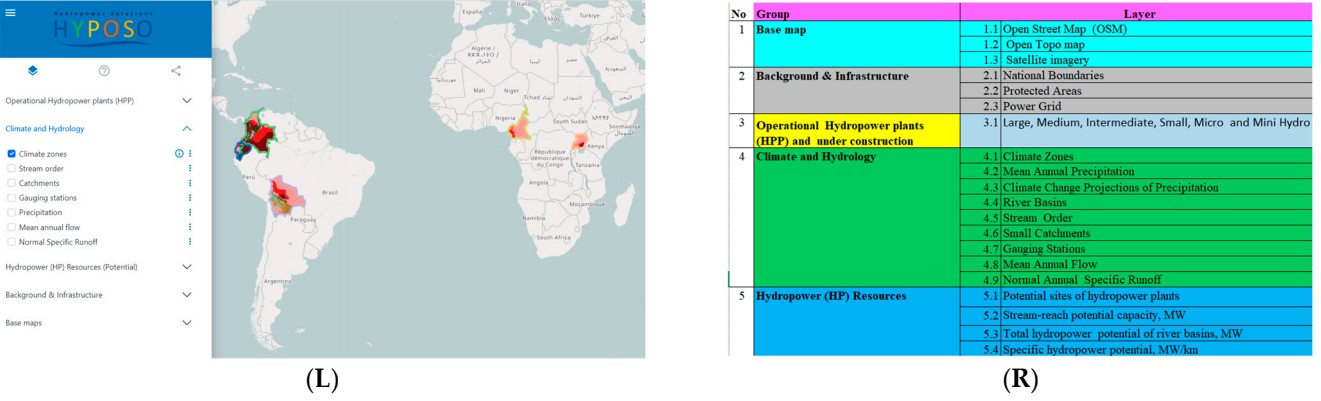

(**L**)      (**R**)

**Figure 2.** Front page of the HYPOSO web map viewer (**L**) and the structure of the layers (**R**).

The river basin layer represents hydrologic units (regions), water management districts, or large to medium river basins [33]. This information was collected from the national hydrologic institution [34]. Boundaries of the major river basins were rendered by GIS tools. Stream order was used to describe the hierarchy of streams from the top to the bottom of a catchment according to the Strahler system [35]. The highest one was attributed to the Albert and Victoria Nile. In this system, the smallest headwater tributaries are called first-order streams. Where two first-order streams meet, a second-order stream is created; where two second-order streams meet, a third-order stream is made, and so on. When considering this order, a general insight into the flow size of a stream to assess its

power capacity on a large scale can be made. In addition, this study evaluated the total hydropower potential according to the Strahler stream ordering system, which is seldom described in quantitative terms in the literature.

In this study, the normal specific runoff (q—river discharge per square kilometre—L/s·km$^2$) was derived from mean annual flow, and maps were produced in a colour palette to characterise long-term mean annual river flow. It was used to compute stream flow for ungauged basins. Long-term mean annual flow allows for calculating the country's hydropower potential (see formula 1). However, the hydrometric network must be sufficiently dense, the length of the records long enough without data gaps, and the data quality assured [36]. The study used the historical mean annual flow series of river gauging stations (G.S.). They were collected from various sources, mainly from the Directorate of Water Resources Management-DWRM in Uganda (Table 3).

**Table 3.** River gauging stations and mean annual flow data sources in Uganda.

| Source, Reference | Number of Gauging Stations (G.S.) | Description |
|---|---|---|
| Directorate of Water Resources Management (DWRM) | 69 | The record period covers from 1947 to 2020. with flow data length between 5 and 73 y. and an average of 46 y. Most G.S. (51) flow data series comprised 35 to 45 y. |
| Global Runoff Data Centre (GRDC) [37] | 12 | The record period covers from 1946 to 1982. with flow data lengths between 3 and 25 y. |
| SIEREM [38] | 63 | A list of G.S. with coordinates is presented but only six G.S. with flow data. The record period covers from 1976 to 1979. |
| DWRM [34] | 71 | A general description of G.S. operating (71) or closed (55) in major river basins. The record period covers from 1978 to 2014. Neither G.S. catchment areas nor coordinates are given for a number of G.S. Monthly flow statistics are summarised. |

Gauging stations operating in Uganda with concise information (coordinates, catchment areas, length of records, and other relevant information) can be observed in the Hyposo map (Figure 3).

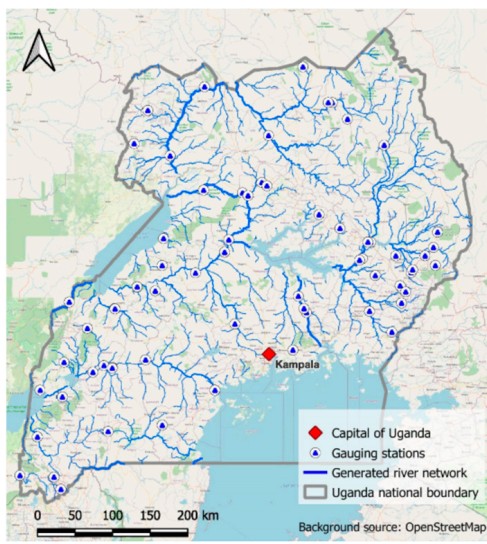

**Figure 3.** Stream gauging stations (G.S.) network in Uganda.

The availability of sufficiently good-quality data underpins all aspects of hydrology, from research to water resource assessment through to a wide range of operational applications [39]. The complicated hydrography and hydrologic regime of the streams in the country was a big challenge. In addition, the scarcity of flow data representing river network, gaps in missing observations, wrong G.S. location coordinates, catchment areas, erroneous flow data and subsequent adjustments, including a lack of ability to access yearbooks of hydrological data in the country, were the main issues related to the collected dataset.

To crosscheck for the information on the gauging stations and consistency of flow data, a number of research papers, hydrological and river basin management reports available at open sources were used [18,40]. Many authors of hydrological studies point out shortcomings related to the placement of the gauges in the river reach concerning stream hydraulics, their representativeness and the ability to measure the discharge adequately. For instance, only approximately one-third could be appropriately used after carefully screening some 16 operating or closed G.S. in the Lake Kyoga basin [41]. Studies in the Mpanga River catchment highlighted a high-level uncertainty in collected flow data records [42].

A comprehensive review of the hydrological data collected in 1978–2014 emphasises the need for high-quality water data [34]. However, according to the Ministry of Water and Environment of Uganda (2019), data quality has declined since the early 2000s. It should be noted that the poor quality of the gauged flow records in a number of western African countries prevented a web-based hydropower atlas developer from using them to assess hydropower potential [43]. Instead, a simplified water balance model was applied to estimate the river flow. Taking the above issues into consideration, hydrological modelling will face a number of significant challenges [36]. Finally, the reliability and quality of input data series for this study were judged sufficient only at a minimal level.

Two types of specific runoff maps were compiled: (a) for the entire Ugandan territory when the hydrographic network (excluding the Nile) features are not considered; (b) by river basin. A total of 48 river basins were identified, a priori regarded as homogeneous for the generation of river flow. However, not all river basins were covered by gauging stations, and there were also unrepresentative gauges.

A geospatial interpolation method was applied to produce the specific runoff maps. Many sources describe geospatial interpolation methods, to name just a few [44,45]. But there is no single general method for data interpolation. The selection of the interpolation method is usually based on several criteria, i.e., the actual amount of data, the required level of accuracy and the time and/or available computer resources. Considering the quantity and quality of the available flow data, the inverse distance weighting (IDW) method was chosen for interpolation [46,47]. In the IDW method, weights are assigned to the values of the starting points, inversely proportional to the distance to the point under investigation. Specific runoff distribution maps were created using the point layer of the stream gauging stations (using specific runoff values) and GIS Spatial Analyst Interpolation-IDW. However, beforehand, due to unjustified high values in some small catchments, manual adjustment was required to ensure accuracy.

The river network and sub-catchment GIS layers were created with relevant attributes, displaying the hydropower potential. To delineate the river network and subbasins (small catchments), the MERIT Hydro digital elevation model (DEM) represented the terrain elevations at a 3 arcseconds resolution (~90 m at the equator) was used [48]. This DEM was hydrologically conditioned, and a well-known gravitation-based model was applied to delineate stream networks and sub-basins [49]. ESRI ArcGIS Pro with the ArcHydro toolset was employed for data processing.

Hydropower potential was calculated based on the longitudinal river profile between two successive confluences. It has been proven that when a 200 km-long river is divided into 5 or 6 sections, the potential energy found differs by less than 10% from that obtained when divided into 30 or 40 sections [50]. Therefore, a very detailed splitting of the

stream into short segments does not significantly increase the accuracy of the potential energy estimation.

The gross hydraulic head (in our case, the height difference) and flow for the segmented rivers (some 4560 stream-reaches) were determined, and the hydropower capacity or potential was calculated based on this formula:

$$P = c \cdot H \cdot (Q_u + Q_d)/2 \tag{1}$$

where P is the hydropower potential (MW). c is a constant for considering unit conversion and approximate overall plant efficiency, including hydraulic losses [51,52]. This study assumed the following value: c = 8.5/1000. H is the elevation difference from the start to the end of a river reach (m). Q is the long-term mean annual discharge upstream (start) and downstream (end) of a river reach ($m^3$/s).

For small hydropower development, the most important task is to identify stream catchment (sub-basin) boundaries and flow-contributing areas to derive hydrologic metrics. For this country, small sub-basin areas were produced according to the DEM data. A threshold of 25 $km^2$ was used to define streams and sub-basins, which means that small streams with a flow-contributing area under 25 $km^2$ were not considered in this project.

Validation of longitudinal river profiles of the channel elevation along the river course was performed. These profiles were created from the DEM data and compared with those produced from topographic maps and alternative GIS assessments [18]. Selected rivers belonged to the Lake Kyoga river basin, a water management zone or major river basin, including Ririrma, a tributary of the Mpologoma river.

The total hydropower potential in Uganda was assessed using the GIS tools, then the Nile potential was excluded, and small hydro resources were distinguished up to 10 and 20 MW. They can be considered as technically feasible potential. After that, the river reaches were screened out excluding the rivers that fall into the country's protected areas. Protected areas are dedicated to preserving the biological diversity and natural, recreational, and cultural resources that are managed through legal means. The World Database on Protected Areas (WDPA) was examined [53], providing the environmentally compliant hydropower potential [54,55]. However, this does not mean that SHP development is completely banned in the designated areas. There are plenty of examples worldwide of SHP successfully operating in these areas. It depends on the level of environmental sensitivity of the protected area and social-economic factors.

Apart from the operational HPPs and a number of potential site locations available at MEMD [25], all of the data published in the GIS layers were based on modelling results. The latter considers protected areas, the proximity of the grid, and settlements. Although the modelled estimates do not represent the actual numbers feasible for engineering design, they provide the basis for follow-up pre-feasibility or feasibility studies. However, georeferenced points of potential sites do not differentiate between SHP intake and powerhouse but rather indicate the best-suited river reaches for SHP development.

A descriptive statistic that quantitatively describes or summarises features from collected or generated datasets was employed. The best fitted frequency distributions were selected. GIS modelling procedures are not detailed in this article.

## 3. Results and Discussion

### 3.1. Validation of Longitudinal Stream Profiles

DEM accuracy impacts the magnitude of hydropower potential, particularly the river channel slope or a drop in river channel elevation. The SRTM (Shuttle Radar Topography Mission) [56] was designed with specific mapping accuracy thresholds to help ensure a consistent and accurate global topographic dataset. In the literature, there is no clear consensus on assessing the accuracy of SRTM DEM. It depends on geographic region, the individual variability of topography, and land cover conditions [57].

For Africa, the SRTM dataset has an average horizontal error of 11.9 m and an average absolute vertical (elevation) error of 5.6 m [58]. MERIT DEM estimates a vertical accuracy

of 12 m [48,59]. According to a review conducted by [57] the SRTM DEM, with a spatial resolution of 30 m, has a reported accuracy of 16 m. The vertical accuracy has also been reported to be <9 m and 4.31 m in mountain regions.

The longitudinal stream profiles extracted from the HYPOSO DEM [19], GIS DEM [18], and the topographic map with contour lines (TOPO) were compared (Figure 4). Their key descriptive statistics are shown in Table 4.

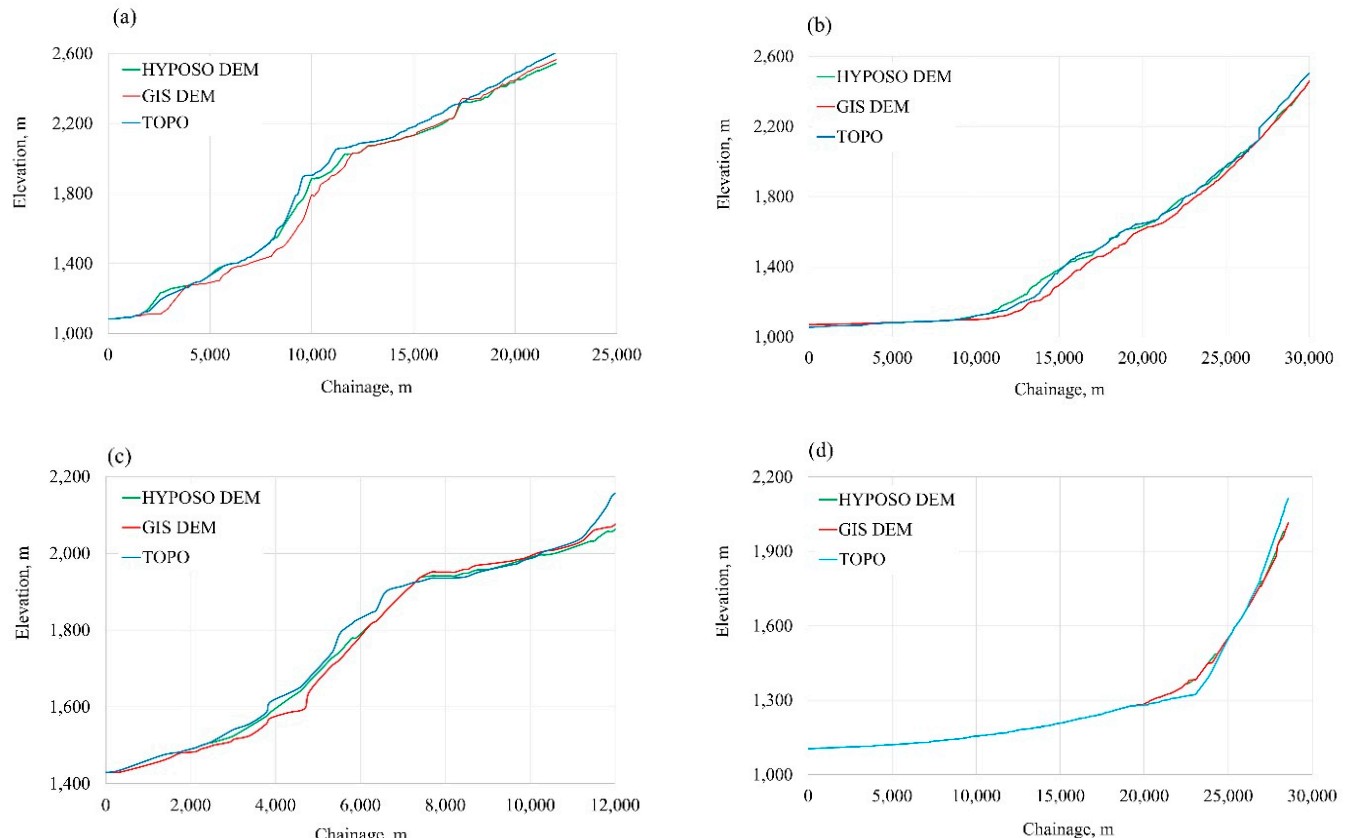

**Figure 4.** Comparison of longitudinal stream profiles of the datasets of HYPOSO DEM [19], GIS DEM, [18] and the reference profile (topographic map with contour lines). (**a**) Sipi, (**b**) Simu, (**c**) Ririrma and (**d**) Sironko.

**Table 4.** Results of the statistical analysis of the compared longitudinal stream profiles from DEMs to the topological map.

| Statistics | Stream Name | | | | | | | |
| --- | --- | --- | --- | --- | --- | --- | --- | --- |
| | Sipi | | Simu | | Ririrma | | Sironko | |
| | HYPOSO | GIS | HYPOSO | GIS | HYPOSO | GIS | HYPOSO | GIS |
| Sample size | 110 | 110 | 168 | 168 | 54 | 54 | 141 | 141 |
| RMSE, m | 40.46 | 67.36 | 23.73 | 40.74 | 27.69 | 32.39 | 23.49 | 24.73 |
| Standard deviation of errors, m | 30.43 | 46.41 | 23.76 | 29.16 | 24.07 | 28.08 | 23.58 | 24.82 |
| Determination coefficient | 0.998 | 0.991 | 0.998 | 0.996 | 0.989 | 0.958 | 0.992 | 0.992 |

As one can see from Table 4, the statistical estimates were good for the steep topography areas of Uganda, while the evaluation fell short of engineering standards in the flat landscape. GIS applications usually rely on the combination of layers with various

temporal–spatial resolutions and geographic projections. Spatial accuracy is therefore likely to affect the result quality and accuracy adversely [11].

### 3.2. Generated Rivers and Small Sub-Basins

The river and stream network was generated (Figure 5a), and stream-reach capacities were calculated. A total of 4560 river reaches were generated, of which 362 had a very low elevation drop. These reaches cannot be considered effective for hydropower, as a dam impounding a river or stream will result in a lengthy backwater stretch, i.e., the slope of the channel was slight (<30–40 cm/km), or their length was short with a slightly higher slope.

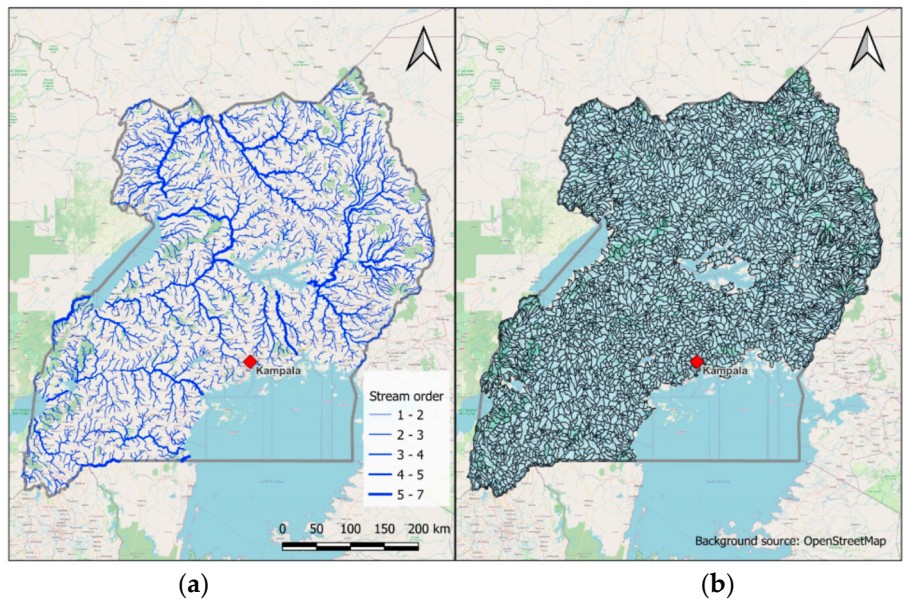

(**a**)                                                   (**b**)

**Figure 5.** Generated rivers and streams with the Strahler stream order highlighted (**a**) and boundaries of small sub-basins (**b**) of Uganda.

For small hydropower development, the most important task is to identify stream catchment boundaries and their areas to derive hydrologic metrics. Small sub-basins (catchments) were delineated from DEM (Figure 5b), and their flow-contributing areas were determined. A histogram or frequency distribution diagram of the areas of the generated sub-basins (A, km$^2$) for the entire country was produced (Figure 6).

The geographic area of Uganda is 241,038 km$^2$, in which 4664 sub-basins were identified. The mean sub-basin area was 43.2 km$^2$, the maximum area was 332.5 km$^2$, and the standard deviation of the dataset was 34.3 km$^2$ (Table 5).

**Table 5.** Basic statistics of the areas of generated small sub-basins in Uganda.

| Small Sub-Basins Area, km$^2$ | | | |
|---|---|---|---|
| Mean | 43.2 | Sample variance | 1176.9 |
| Standard error | 0.5 | Minimum | 0.1 |
| Median | 36.1 | Maximum | 332.5 |
| Standard deviation | 34.3 | Sample size | 4664 |

Once a flow-contributing area of a sub-basin is determined, it becomes easier to derive hydrological metrics, e.g., the mean flow, from the available specific runoff maps (L/s or m$^3$/s km$^2$) for a prospective hydro scheme.

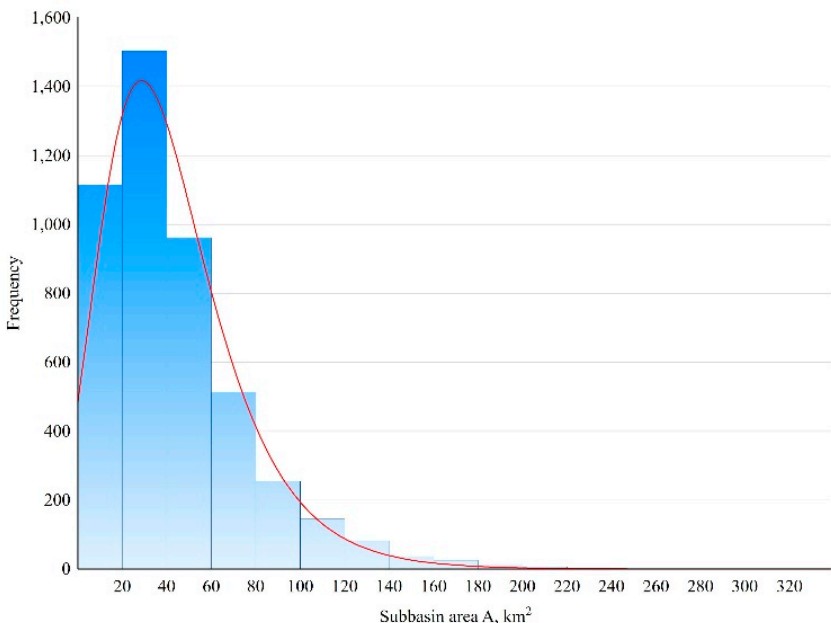

**Figure 6.** Frequency distribution of the generated small sub-basin areas in Uganda. The curve indicates the theoretical extreme (Type 1) distribution density function.

### 3.3. Specific Runoff

There is no available detailed mean annual river flow or specific runoff (q) map for Uganda. Uganda's only available specific runoff data consisted of the specific runoff map developed some twenty-five years ago based on flow records between 1950 and 1967 [60]. However, the reliability of this map was considered not satisfactory [2]. A crosscheck implemented with gauged specific runoff estimates for small river basins (areas between 70 and 660 km$^2$) revealed that their differences ranged from 3 to 44 times.

Based on the methodology mentioned above, two specific runoff maps were produced using the IDW interpolation method (Figure 7). As can be seen, at first glance, they look similar, but the map for the entire hydrographic network (left) is more detailed in terms of the magnitude of the specific runoff. When carrying it out, the IDW employed unjustified high values of small catchments and disproportionately increased specific runoff in major river basins. To elaborate a specific runoff map for the separated river basins, these extreme values reflecting local conditions were manually adjusted, or smoothing was performed. It was used for determining stream-reach hydropower potential. In both cases, the insufficient coverage of the basins by gauging stations was a shortcoming of the interpolation method.

The country exhibits a high variation in the specific runoff. High specific runoff values are only observed in southwestern Uganda, the West Nile, and the Mt. Elon region. In small catchments, they can reach up to 40 L/s·km$^2$. Very low values are reported in the Lake Kyoga area, Katonga and Bukora catchments, and Albert Nile valley. Based on the developed specific runoff digital map, the natural mean annual runoff volume for the country's geographic area was estimated to be 44.31 bln. m$^3$. It should be noted that an alternative assessment of this volume found 40.8 bln. m$^3$ [11]. The difference can be attributed to the difference in the initial hydrological data, their period of observation, and data accuracy including the spatial location of the gauging stations.

Validation of the specific runoff map is shown below (Figure 8 and Table 6). The best way was to compare the mean annual flow gauged in the G.S. and calculated from the specific runoff map. They show that a satisfactory correlation result was obtained.

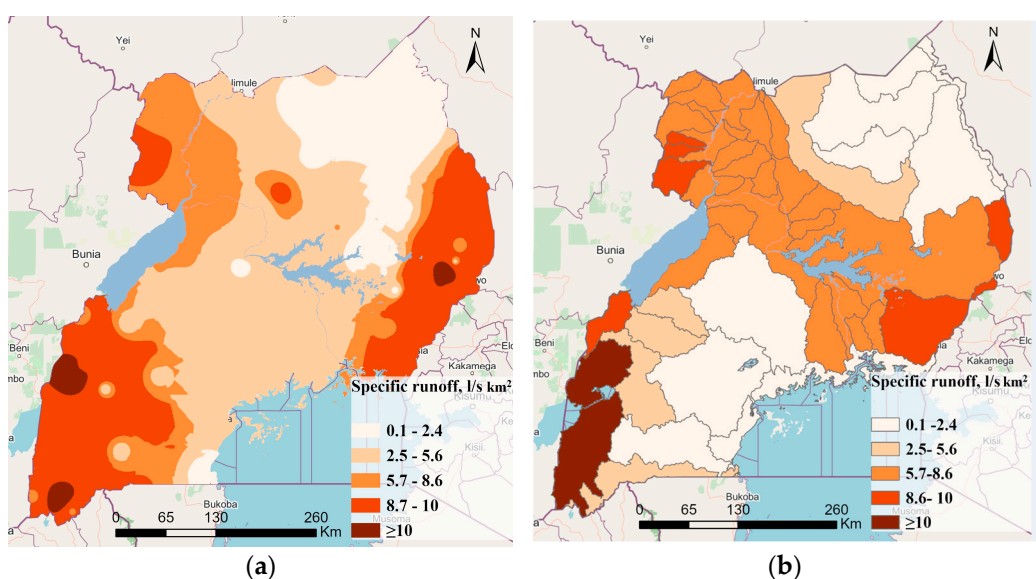

**Figure 7.** Specific runoff map of Uganda. Geospatial interpolation was performed for the whole country's hydrographic area (**a**) and main river basins (**b**).

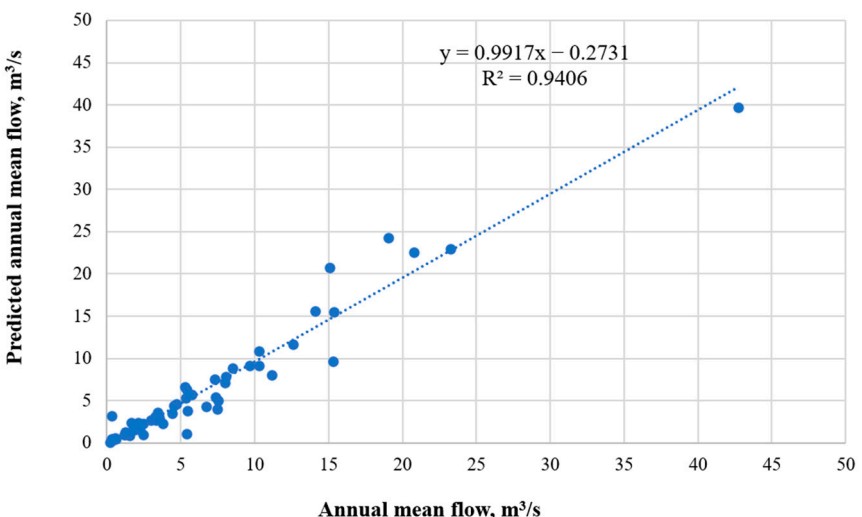

**Figure 8.** Relationship of mean annual flow at gauging stations with modelled mean annual flow derived from the specific runoff map (Figure 7).

**Table 6.** Summary of regression analysis (gauged flow data and modelled from the specific runoff map).

| Mean Annual River Flow, m³/s | Sample Size | Mean | Standard Deviation | Standard Error | Median | Sample Variance | Min | Max |
|---|---|---|---|---|---|---|---|---|
| Gauging stations | 55 | 6.81 | 7.35 | 0.99 | 4.71 | 54.04 | 0.20 | 42.70 |
| Predicted [1] | 55 | 6.48 | 7.51 | 1.01 | 3.79 | 56.50 | 0.12 | 39.70 |

[1] Modelled from the specific runoff map (Figure 7b).

### 3.4. Hydropower Potential

#### 3.4.1. Stream-Reach Capacity

The final product of the hydropower assessment was a GIS stream vector layer with attributes containing generated stream/river segment length (km), slope (m/km), segment head height (m), upstream and downstream drainage areas (km²) and corresponding flow

(m³/s), potential capacity in MW and MW/km, Strahler stream order, environmental sensitivity, and grid proximity (Figure 9).

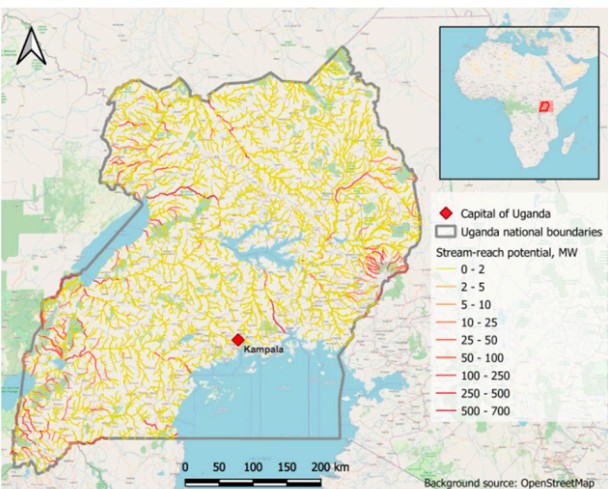

**Figure 9.** Stream-reach potential capacity (MW) of Uganda.

Most authors indicate that the Strahler stream order must be at least third to ensure the sufficient availability of flow in streams for effective hydropower production [10,11,61,62]. In other words, prior studies evaluating water power potential in first- and second-order streams ignored their contribution. The available geospatial data made it possible to assess the total potential according to the Strahler stream order of rivers and their total number (Table 7) and investigate the power capacity of individual rivers corresponding to such a category (Figure 10).

**Table 7.** Hydropower potential according to the Strahler stream order.

| Stream Order | Number of Streams and Rivers | Total Power Capacity, MW | Percentage | Average Power Capacity Per Stream or River, MW |
|---|---|---|---|---|
| 1 | 2373 | 614.6 | 8.8 | 0.3 |
| 2 | 516 | 608.4 | 8.7 | 1.2 |
| 3 | 117 | 341.7 | 4.9 | 2.9 |
| 4 | 29 | 275.7 | 3.9 | 9.5 |
| 5 | 7 | 86.7 | 1.2 | 12.4 |
| 6 | 5 | 226.9 | 3.2 | 45.4 |
| 7 | 2 [1] | 4831.4 | 69.2 | 1610.5 |
| Total: | 3050 | 6985.4 | 100.0 | 2.3 |

[1] Victoria and Albert Nile.

As can be seen from Table 7, the total hydropower potential of first- and second-order streams is quite impressive if the Nile is not considered. Its share is close to 18% of the total. However, the average power capacity per stream is relatively low (0.26–1.23 MW). But there might be exceptions for some steep-topography creeks, which generally have a low flow rate, but significant water drops in elevation and, simultaneously, have a high power capacity. This is especially true for their single reaches that show quite powerful energy intensity.

Rivers in Uganda belong to nine major river basins. The Victoria Nile river basin has the highest total capacity of about 4404 MW. The Lake Edward, Lake Victoria, and Lake Kyoga basins have a total capacity of about 780 and 635 and 296 MW, respectively. The

Albert Nile and Lake Albert basins have a total stream capacity of 296 and 270 MW. The total capacity of the Aswa river basin is about 218 MW. The Turkwei and Kidepo river basins have the lowest total capacity of 53 and 0.55 MW (Figure 11). In the modelling, only the rivers within the country borders were considered, but the inflow from neighbouring regions was also added to the total flow values.

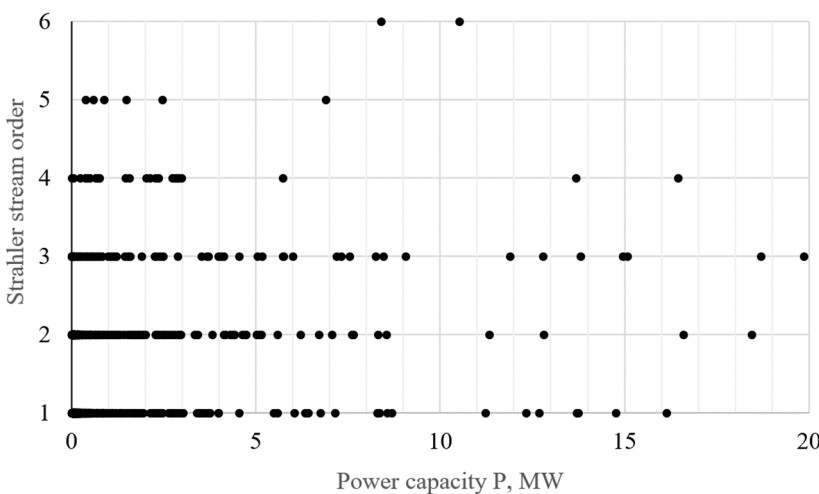

**Figure 10.** Individual rivers' power capacity (<20 MW) following the Strahler stream ordering (Nile is excluded).

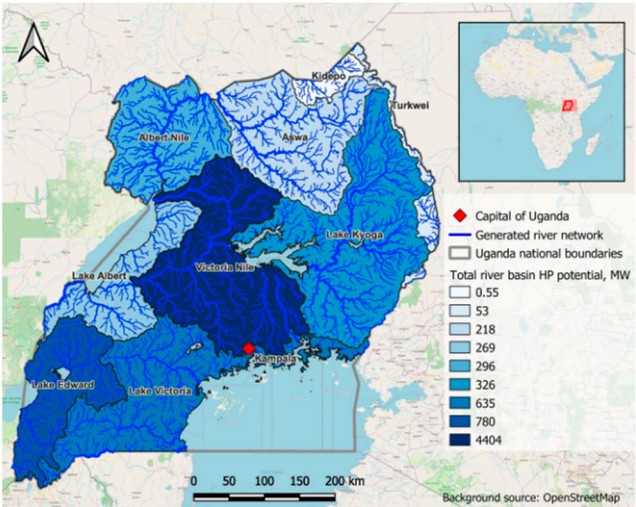

**Figure 11.** Total hydropower potential (MW) of the major river basins.

The following cases were distinguished from assessing the total hydropower potential (Table 8):

(a) All the country's small and medium-sized rivers, including the Nile;
(b) The Nile;
(c) All small and medium-sized rivers in the country, excluding the Nile;
(d) Ditto, excluding the Nile and protected areas;
(e) Ditto, excluding the Nile and stream-reach potential capacity of more than 20 MW (national SHP capacity limit);
(f) Ditto, excluding the Nile and stream-reach potential capacity of more than 10 MW (European SHP standard).

**Table 8.** Basic statistics of hydropower potential (MW) in Uganda.

| No | Metric | All Rivers and Streams, Including the Nile (a) | The Nile (b) | Small and Medium-Sized Rivers(the Nile is Excluded) | | | |
|---|---|---|---|---|---|---|---|
| | | | | Including Protected Areas (c) | Protected Areas Excluded (d) | Including Protected Areas | |
| | | | | | | $p < 20$ MW (e) | $p < 10$ MW (f) |
| 1 | Mean | 4.81 | 185.82 | 1.48 | 1.04 | 1.11 | 0.86 |
| 2 | Standard Error | 1 | 42.59 | 0.11 | 0.09 | 0.06 | 0.04 |
| 3 | Median | 0.36 | 78.98 | 0.35 | 0.30 | 0.34 | 0.33 |
| 4 | Standard Deviation | 37.95 | 217.16 | 4.20 | 2.72 | 2.31 | 1.42 |
| 5 | Variance | 1440 | 47,160 | 17.61 | 7.41 | 5.34 | 2.01 |
| 6 | Range | 673.52 | 664.53 | 58.19 | 41.77 | 19.51 | 9.71 |
| 7 | Minimum | 0.1 | 9.09 | 0.1 | 0.1 | 0.1 | 0.1 |
| 8 | Maximum | 673.62 | 673.62 | 58.29 | 41.87 | 19.61 | 9.81 |
| 9 | Sum | 6917 | 4831 | 2086 | 1033 | 1549 | 1181 |
| 10 | Sample size | 1439 | 26 | 1413 | 992 | 1396 | 1370 |

Low-energy intensity ($p < 0.1$MW) stream reaches (some 3120 out of 4560 segments) were eliminated. Their total power capacity is insignificant compared to the total country's hydropower potential (less than 1%).

(a) The total stream-reach capacity potential for the country's hydrographic area, including the Nile, was identified as 6917 MW (1439 stream reaches), which can be regarded as the technically feasible potential. Of these, 4170 MW of rivers' capacity potential (and streams) falls within protected areas. This is about 60% of the total potential. Rivers that partially fall into protected areas add another 260 MW. Partially falling into a protected area was defined as up to ~30% of the river's length occupying a protected area. The mean capacity of a stream segment was 4.81 MW, with a maximum of 673.62 MW (Table 8). Other estimates for the total hydropower potential differ considerably, e.g., 4137–4500 MW [5,29,30]. This is roughly one and a half times lower. The *World Atlas of Hydropower & Dams* [26] suggests that the capacity is 6950 MW. In contrast, a GIS study using the RhydroSearch application showed that the total hydropower potential estimates are at least three times higher, between 22,190 and 24,622 MW [3,27];

(b) The Nile. In this case, the total stream-reach capacity potential amounts to 4831 MW, with a mean of 185.82 MW (population size is 26 out of 96 river segments). More than 1410 MW of capacity potential is currently installed or under construction [19]. As a result, the remaining potential amounts to 3421 MW;

(c) The total stream-reach capacity potential of small and medium-sized rivers (the Nile is excluded) amounts to 2086 MW;

(d) If the protected areas are screened out of the reaches mentioned above, the total capacity diminishes by nearly half to 1033 MW;

(e) If the stream-reach capacity is considered only at less than 20 MW, then the total stream-reach capacity potential decreases to 1549 MW. Their individual sites' power capacity frequency distribution is illustrated in Figure 12. More than 160 MW are currently installed in SHPs < 20 MW [5,23]. As a result, the remaining potential amounts to some 1389 MW, or approximately 5500 GWh/year. The World Small Hydropower Development Report (WSHDR) for Uganda [32] provides a relatively low estimate of 258 MW. No detailed information was given in this report on how this potential was determined;

(f) If the stream-reach capacity is considered only at less than 10 MW, then the total stream-reach capacity potential decreases to 1181 MW (sample size is 1370). More than 100 MW are currently installed in SHPs <10 MW [19,23]. As a result, the remaining

potential amounts to 1081 MW. The WSHDR estimate for this SHP capacity limit was relatively low at 200 MW [32]. A very small technical SHP potential, taking into account environmental limitations, was estimated at 49.8 MW in 43 sites in Uganda [11].

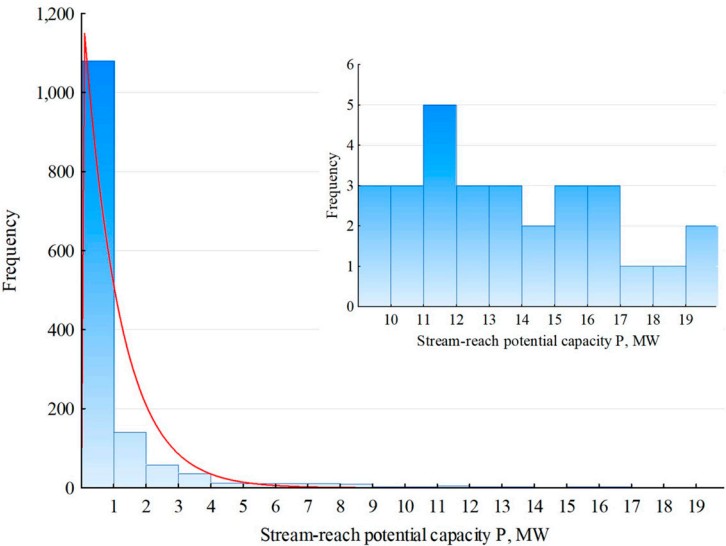

**Figure 12.** Frequency distribution of stream-reach capacities up to 20 MW with fitted theoretical exponential distribution density function.

3.4.2. Potential Hydropower Sites

A dataset of more than 500 new potential areas for hydropower development, including 100 points from MEMD [25] and those generated by GIS modelling (the Nile is excluded), is illustrated in Figure 13. Their key metrics are illustrated in Figure 14a. The aim was to analyse stream-reaches with high energy density and to compile and spatially join the energy potential of stream-reaches with technical (e.g., grid network, energy demand concentration points) and non-technical information (network of protected areas).

The following basic features can be Identified on the HYPOSO map viewer:

- Site type (e.g., run-of-the-river, reservoir, off-grid, or central grid);
- Address, stream or river name, basin (hydrologic unit or water management district name), coordinates (longitude and latitude);
- Approximate capacity (MW), flow ($m^3/s$), and head (m);
- Environmental sensitivity (e.g., protected areas);
- Any opportunities for development (e.g., prior studies).

The capacity frequency distribution pattern of the potential sites for development (the head and capacity above 5 m and 0.1MW, respectively) is illustrated in Figure 14b.

The total capacity of the 485 identified sites amounted to 1217.5 MW, with a mean, maximum, and minimum, respectively, of 2.51, 0.18, and 19.94 MW (Table 9). A total of 162 sites, of which the total capacity was 432 MW, were located within protected areas.

**Table 9.** Descriptive statistics of potential site capacities ($p < 20$ MW).

| Capacity MW | | | | | | | |
|---|---|---|---|---|---|---|---|
| Mean | 2.51 | Mode | 1.37 | Range | 19.75 | Sum | 1217.5 |
| Standard Error | 0.14 | Standard Deviation | 3.19 | Min | 0.18 | Sample size | 485 |
| Median | 1.2 | Sample Variance | 10.15 | Max | 19.94 | | |

It should be noted that GIS modelling produced an energy-efficient stream reach between two adjacent tributaries, i.e., the maximum possible capacity of a natural stream segment. The optimal ones were selected based on the highest potential capacity, distance to the nearest settlements, and proximity to the grid. The environmental sensitivity was also specified. Moreover, the capacity of a natural river section can be increased artificially. For instance, if a derivation scheme is applied, an increase in the head is usually possible. A reconnaissance study is necessary for the precise locations of the SHP.

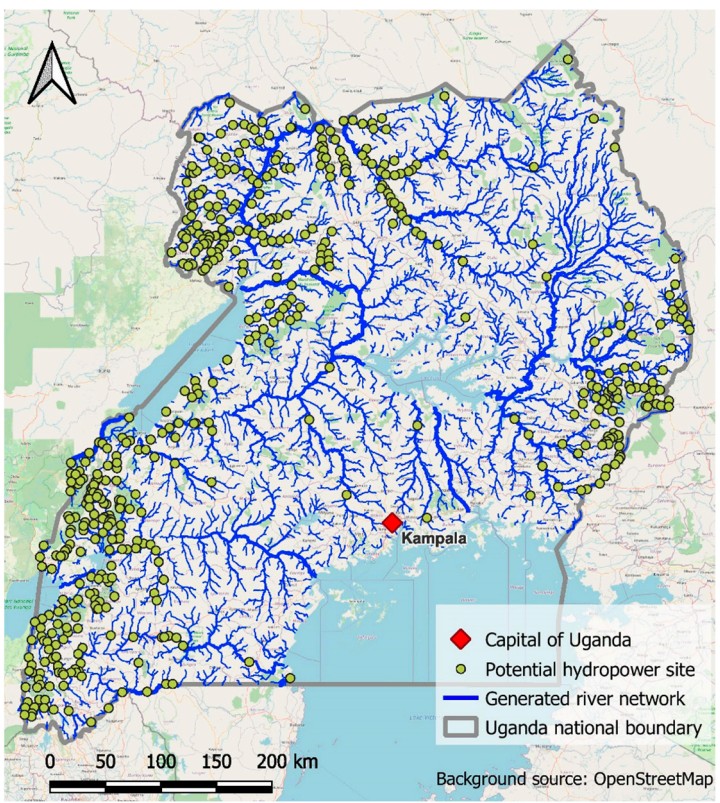

**Figure 13.** Potential georeferenced hydropower sites in Uganda.

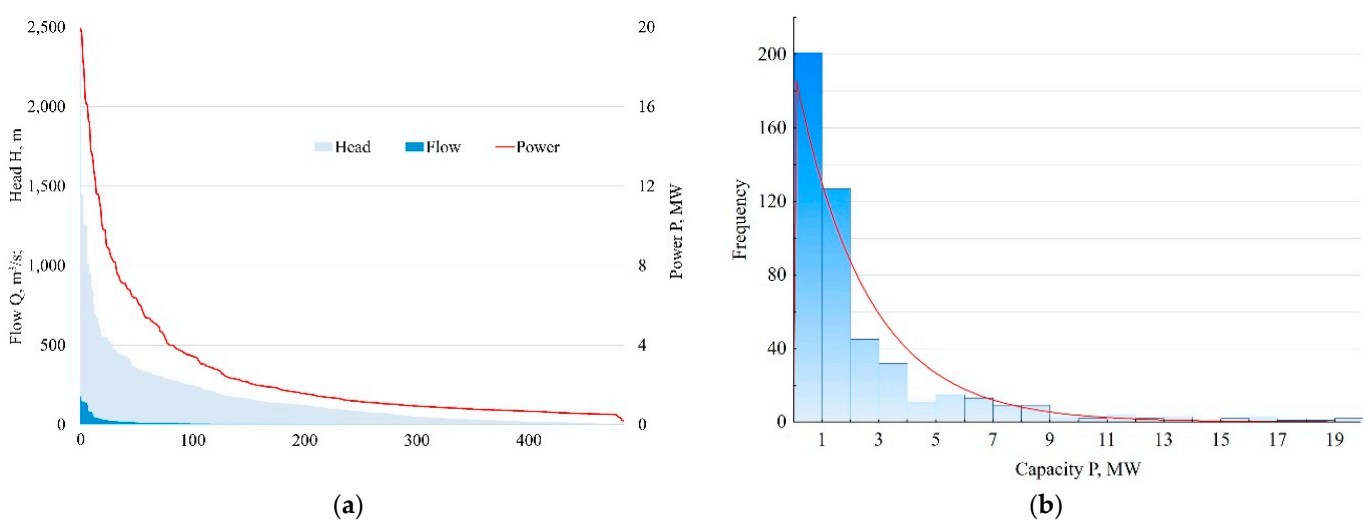

**Figure 14.** Key characteristics of the potential hydropower sites at large (**a**) and their frequency distribution of capacities ($p < 20$ MW) with a fitted theoretical exponential distribution density function (**b**).

## 4. Conclusions

1. This article presents a summary of Uganda's hydropower characteristics that were extracted from the HYPOSO map and processed. The raw data are freely available. The latter provides practical opportunities to examine the prospective hydro schemes of specific areas in much more detail. However, in any case, the modelled estimates do not represent the actual numbers for engineering design; they provide the basis for follow-up studies to proceed with pre-feasibility or feasibility studies.

2. The compiled HYPOSO DEM was validated. Longitudinal comparisons of stream profiles showed that the compliance was satisfactory. However, this analysis would not be valid in flat topographic areas, as the accuracy may be unsatisfactory.

3. The boundaries of small sub-basins were identified, and the sizes of areas contributing to runoff were determined. This is core information that is needed in developing SHPs to reveal a first estimate of the river flow based on the normal specific runoff digital map.

4. Uganda's hydropower potential was determined, and its values were compared with prior estimates. Notable discrepancies were highlighted, and the reasons for them were briefly discussed.

5. A dataset of potential SHP site locations for hydropower exploitation was compiled, covering some 500 points for Uganda and taking into account expected capacity, protected areas, the proximity of the grid, settlements, and concentration points of energy demand. Screening out of protected areas does not mean that hydropower development is completely excluded. The level of environmental sensitivity, legislation and social-economic factors should be taken into account.

6. A concise statistical analysis of the hydropower potential datasets in consideration is presented. These summaries will be necessary for decision-makers to foster SHP development in this country.

**Author Contributions:** Conceptualization, P.P. and L.J.; methodology, G.V.; validation, P.P. and L.J.; formal analysis, L.J.; investigation, P.P.; data curation, A.K.; writing—original draft preparation, P.P.; writing—review and editing, G.V.; visualization, A.K.; supervision, P.P. All authors have read and agreed to the published version of the manuscript.

**Funding:** This research was funded by the European Commission (EC) and the European Climate, Infrastructure and Environment Executive Agency (CINEA), grant number 857851.

**Acknowledgments:** The authors thank the European Commission and CINEA for supporting the HYPOSO (Hydropower solutions for developing and emerging countries) project. This paper is based on the findings of the ongoing work package WP3 "Framework Analysis and Research Needs", task 3.3, "Mapping potential hydropower hotspots in the target countries". The HYPOSO web mapping was carried out by A. Balčiūnas., A. Dumbrauskas performed part of the GIS modelling. The authors are grateful to D.M. Nabutsabi, of the Ugandan Hydropower Association, for their collaboration in the project.

**Conflicts of Interest:** The authors declare no conflict of interest. The funders had no role in the design of the study; in the collection, analyses, or interpretation of data; in the writing of the manuscript; or in the decision to publish the results.

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
