# Peer review of "Small Hydropower Assessment of Uganda Based on Multisource Geospatial Data"

_water, doi:10.3390/w15112051_

Round 1

Reviewer 1 Report

The writing is well tailored, and the explanations are often reasonable on multi-geospatial data issue. However, sometimes, at some points, the submitted research leaves something to be desired. Nevertheless, the presented manuscript could be recommended for the publication after a minor revision and amendments following these next comments:

1.      In abstract, author should avoid the use of very long sentences and abbreviations phrases.

2.      The contents of Introduction (especially 1.2) are too long and they need to be shortened.

3.      The main objectives and novelty aspects in this study should be highlighted clearly within the introduction section.

4.      Figure 1 is not clear and it needs to be replaced.

5.      In Table 1, authors should state what does Ao mean.

6.      In line 112, (SHP) should be used only because authors have explain what does (SHP) mean in line 81.

7.      References should be revised to ensure that volume, and start-ending pages are provided, whenever possible.

Author Response

We would like to thank the Reviewer and Editor for their thoughtful review of the manuscript. They raise important issues, and their inputs are constructive for improving the manuscript. We agree with almost comments and have revised our manuscript accordingly. Two files (docx) of our manuscript are attached: one file with "track changes", which highlights our revisions and the other file – "clean", without any modifications.

Point 1: In the abstract, the author should avoid the use of very long sentences and abbreviations phrases.

Response 1: Accepted. The abstract was revised.

Point 2: The contents of the Introduction (especially 1.2) are too long, and they need to be shortened.

Response 2:  Accepted. Section 1.2 was shortened.  

Point 3: The main objectives and novelty aspects in this study should be highlighted clearly within the introduction section.

Response 3: Accepted. The text was complimented.

Point 4: Figure 1 is not clear and it needs to be replaced.

Response 4: Accepted. Figure 1 was redrawn. Symbology was altered for a better representation of data.

Point 5: In Table 1, authors should state what does Ao mean.

Response 5: It is the abbreviation of the first author's name (Norwegian)

Point 6:  In line 112, (SHP) should be used only because authors have explain what does (SHP) mean in line 81.

Response 6: Accepted and corrected.

Point 7:  References should be revised to ensure that volume, and start-ending pages are provided, whenever possible.

Response 7: Accepted and corrected.

Reviewer 2 Report

Dear authors,

I have only two minor suggestions:

1. Line 155: "The map comprises 20 layers broken into six groups....". In fact, according to Fig. 2, right, there are only 5 groups.

2. Lines 170-171: "....the normal specific runoff (q − river discharge per square kilometre per second − l/s·km2)...". Please, delete "per second" because the discharge is expressed in l/s, and the specific discharge  in l/s·km2.  Thus, I suggest: "....the normal specific runoff (q − river discharge per square kilometre − l/s·km2)...".

Author Response

We would like to thank the Reviewer and Editor for their thoughtful review of the manuscript. They raise important issues, and their inputs are constructive for improving the manuscript. We agree with almost all their comments, and we have entirely revised our manuscript accordingly.  Two files (docx) of our manuscript are attached: one file with "track changes", which highlights our revisions and the other file – "clean", without any revisions.

Point 1:  Line 155: "The map comprises 20 layers broken into six groups....". In fact, according to Fig. 2, right, there are only 5 groups.

Response 1:  Accepted and corrected.

Point 2:  Lines 170-171: "....the normal specific runoff (q − river discharge per square kilometre per second − l/s·km2)...". Please, delete "per second" because the discharge is expressed in l/s, and the specific discharge  in l/s·km2.  Thus, I suggest: "....the normal specific runoff (q − river discharge per square kilometre − l/s·km2)...".

Response 2: Thank you for this rightful comment. Accepted and corrected